# Random-Walk Laplacian for Frequency Analysis in Periodic Graphs

**DOI:** 10.3390/s21041275

**Published:** 2021-02-11

**Authors:** Rachid Boukrab, Alba Pagès-Zamora

**Affiliations:** SPCOM Group, Universitat Politècnica de Catalunya-Barcelona Tech, 08034 Barcelona, Spain; rachid.boukrab@upc.edu

**Keywords:** graph Fourier transform, frequency ordering, random-walk Laplacian, periodic graph

## Abstract

This paper presents the benefits of using the random-walk normalized Laplacian matrix as a graph-shift operator and defines the frequencies of a graph by the eigenvalues of this matrix. A criterion to order these frequencies is proposed based on the Euclidean distance between a graph signal and its shifted version with the transition matrix as shift operator. Further, the frequencies of a periodic graph built through the repeated concatenation of a basic graph are studied. We show that when a graph is replicated, the graph frequency domain is interpolated by an upsampling factor equal to the number of replicas of the basic graph, similarly to the effect of zero-padding in digital signal processing.

## 1. Introduction

A signal can be seen as a function that assigns values to a set of indices, which represent time instants in temporal signals, as, for instance, audio signals, and space coordinates in spatial signals, as in images. A discrete signal is typically obtained after sampling an analog signal generated by a physical phenomenon, like the vibration of the vocal cords or the light reflected from a visual scene. These discrete signals are defined in a Euclidean domain and can be processed using conventional Discrete Signal Processing (DSP) techniques. On the other hand, graph signals [1,2] are signals whose domain is given by a graph that represents pairwise relations between elements. Graph signals arise in applications of very diverse fields that involve a network as in neuroscience, biology, genomics, telecommunications, and economics. For instance, in sensor networks the graph represents the relative positions between the sensors and the graph signal is the data measured by the sensors [3,4]. In brain networks, each node in the graph represents a region of the brain and the edges the functional relations between these regions; graph signals record the activity of the brain regions [5,6]. In general, graphs are irregular non-Euclidean manifolds so that conventional DSP methods render rather useless to analyze and extract information from graph signals.

Graph Signal Processing (GSP) [1,2] is a recently developed framework useful to study graph signals that merges algebraic graph theory with both classical frequency analysis and filtering design tools from DSP. GSP provides methods for the analysis of brain activity or data gathered in social networks for the identification of contours in images [7,8], the compression and distributed processing of graph signals gathered in a sensor network, among others. Moreover, GSP is gaining momentum with the generalization of Convolutional Neural Networks (CNNs) to process graph data using graph filters in the convolutional layers [9,10,11].

With regards to the frequency analysis of a discrete signal, it is widely accepted that the Discrete Fourier Transform (DFT) provides the frequency components of a signal ordered from low-pass to high-pass components. However, the frequency analysis of graph signals and, in particular, the definition and ordering of frequencies of graph signals from low-pass (or smooth) to high-pass modes still causes certain controversy [12]. The Graph Fourier Transform (GFT) generalizes the concept of Fourier transform to graphs and is given by the eigenvectors of the so-called graph-shift operator [13]. Indeed, a graph-shift operator is a matrix that characterizes the evolution of signals across the graph similarly to time shifts in temporal signals. The most commonly used graph-shift operators for graph spectral analysis are the Laplacian matrix [2] and the adjacency matrix [14].

The spectrum of the Laplacian matrix has already been studied in algebraic graph theory, see, e.g., in [15,16,17]. However, graph theory mainly focuses on the construction, analysis, and manipulation of graphs rather than graph signals. In GSP, spectral graph theory is instead used as a tool to define the frequency spectrum of graph signals and the modes of the GFT [2]. The spectrum of the Laplacian matrix renders a measure of smoothness of graph signals as typically the larger the magnitude of an eigenvalue of the Laplacian matrix is, the higher its associated eigenvector varies across the graph. Therefore, the eigenvalues of the Laplacian matrix are often considered the frequencies of the graph, and from now on we will refer to the eigenvalues of a graph-shift operator as the frequencies of the graph, without distinction. With regards to the use of the adjacency matrix for the frequency analysis in graphs, its eigenvalues do not measure the smoothness of graph signals. Therefore, the authors of [1] use the distance between a graph signal and its shifted version using a normalized adjacency matrix to measure the smoothness of graph signals and supports an ordering criterion of the eigenvalues of the adjacency matrix.

In this work, we propose an alternative graph spectral analysis method where the graph frequencies are the eigenvalues of the random-walk Laplacian matrix. Even though the random-walk Laplacian matrix is recently mentioned in [18] as a graph-shift operator, here we present the benefits of using this matrix to define the GFT in terms of its eigenvalues, eigenvectors, and smoothness on the graph and compare it to other Laplacian matrices. Indeed, the criterion to order the frequencies is related to the total variation of the graph transition matrix, which is intimately related to the random-walk Laplacian matrix. In addition to that, we also show that the replication of a graph results in an upsampling in the frequency domain of the graph when the random-walk Laplacian matrix is used for the GFT. This property further supports the use of the random-walk Laplacian to define the Fourier transform in graphs.

Notation (unless otherwise specified): boldface lowercase letters x denote vectors and boldface uppercase letters X denote matrices. A lowercase letter with sub-index $x_{i}$ denotes the *i*th component of a vector and a boldface uppercase letter with a double sub-index $X_{ij}$ denotes the element in row *i* and column *j* of a matrix. Uppercase calligraphic letters X are used to denote sets. $R^{N}$ and $C^{N}$ stand for the *N*-dimensional real and complex vector space, respectively. Finally, $\mathrm{diag}\left(x\right)$ denotes a diagonal matrix with x in its diagonal.

## 2. Graph Spectral Analysis and Smoothness

Given a graph $G=\left(V,E\right)$ with $N=\left|V\right|$ vertices, where V={1,…,N} denotes the set of vertices and E⊆V×V the set of edges, a *graph signal*$x={\left[x_{1},\ldots ,x_{N}\right]}^{T}\in R^{N}$ defined on G is a mapping that assigns to each vertex in the graph a real number [1]. A *graph shift* of signal x is an operation that replaces the current value of each vertex with a linear combination of its current value and the values at the neighboring vertices [19] as follows,
(1)$$y_{i}=\sum_{j\in N_{i}}s_{ij}x_{j}$$
where $N_{i}\subseteq V$ denotes the set of neighbors of vertex i∈V and $\left\{s_{ij}\right\}\;\forall j\in N_{i}$ are the weights of the linear combination. A *graph-shift operator* is defined in [19] as the matrix $S\in R^{N\times N}$ whose entries are the weights $s_{ij}\in R$. The shifted graph signal $y\in R^{N}$ in (Equation 1) can be computed in a compact form as follows,
(2)y=Sx

A graph-shift operator satisfies the following properties:$S_{ij}=s_{ij}\in R$ for i≠j if the edge (i,j)∈E;$S_{ij}=0$ for i≠j if the edge (i,j)∉E; and$S_{ii}\in R\;\forall i\in V$, i.e., the diagonal entries may take any value.

The most common shift operators found in the literature are the adjacency matrix $A\in {\left\{0,1\right\}}^{N\times N}$, defined for unweighted graphs as $A_{ij}=1$ if the edge (i,j)∈E, and $A_{ij}=0$ otherwise, and the Laplacian matrix is defined as
(3)L=D−A
where $D=\mathrm{diag}(d_{1},d_{2},...,d_{N})$ is the degree matrix, a diagonal matrix where $d_{i}=\sum_{j\in N_{i}}A_{ij}$. Another commonly used variant of the Laplacian matrix is the normalized Laplacian matrix, defined as
(4)$$L^{\left(n\right)}:=D^{-\frac{1}{2}}LD^{-\frac{1}{2}}=I-D^{-\frac{1}{2}}AD^{-\frac{1}{2}}$$

Let $S=V\Lambda V^{-1}$ be the eigendecomposition of S, where $V=\;\left[v_{1},...,v_{N}\right]\in R^{N\times N}$ is the matrix whose columns are the eigenvectors of S and $\Lambda =\mathrm{diag}(\lambda_{1},...,\lambda_{N})$ is the diagonal matrix of eigenvalues of S. The GFT of a graph signal $x\in R^{N}$ is defined as $\tilde{x}=V^{-1}x$, and the inverse GFT by $x=V\tilde{x}$ [18]. As $x=\sum_{i=1}^{N}{\tilde{x}}_{i}v_{i}$, each value ${\tilde{x}}_{i}$ weighs the contribution of $v_{i}$, and therefore of the frequency $\lambda_{i}$, to build the graph signal x. In DSP, the measure of smoothness of eigenvectors is straightforward using the DFT, as they are unit complex exponentials whose variation is fully defined by their frequency. However, when it comes to GSP, smoothness is not uniquely defined and heavily depends on the selected shift operator. The works by Shuman et al. [2] and Sandryhaila et al. [1] proposed two different approaches for frequency analysis in GSP. As explained below, the former is based on the Laplacian matrix and the second one on the adjacency matrix.

The measure of the variation of a function is a well-studied problem in the continuous setting through the use of differential operators. Discrete calculus on graphs [20] defines functions on the vertex set V of a graph, such as, e.g., graph signals, and translates the traditional continuous differential operators to discrete differential operators on graphs. The authors of [2,21] use this mathematical framework to obtain a formal measure of smoothness of graph signals based on the Laplacian matrix, using the discrete differential operators described in [22,23,24] as follows. Let H(V) and H(E) be the Hilbert spaces of the real-valued functions defined on the vertices and the edges of a graph $G=\left(V,E\right)$, respectively. The difference operator $\partial_{j}:H\left(V\right)\to H\left(E\right)$ of a graph signal $x\in R^{N}$ along the edge (i,j)∈E is defined as
(5)$$\partial_{j}x_{i}:=\gamma \left(j,i\right)x_{j}-\gamma \left(i,j\right)x_{i}$$
where $\gamma :E\to R^{+}$ is a real function on the edges. Then, the graph gradient of x at vertex i∈V is defined as
(6)$$\nabla x_{i}:=\left[\partial_{j}x_{i}\;:\;j\in N_{i}\right]$$

The local variation at vertex *i*, which is a measure of the local smoothness of x at vertex *i*, is defined as the norm of the graph gradient
(7)$$∥\nabla x_{i}∥=\sqrt{\sum_{j\in N_{i}}{\left(\partial_{j}x_{i}\right)}^{2}}$$

The *p*-Dirichlet form provides a global measure of the smoothness of a function, and for a graph signal x becomes
(8)$$S_{p}\left(x\right)=\frac{1}{p}\sum_{i=1}^{N}{∥\nabla x_{i}∥}^{p}$$

Different smoothness measures $S_{p}\left(x\right)$ arise for different values of γ(i,j) and *p*. For instance, when $\gamma \left(i,j\right)=\sqrt{A_{ij}}$, $S_{1}\left(x\right)$ is the total variation of the signal and $S_{2}\left(x\right)=x^{T}Lx$; if $\gamma \left(i,j\right)=\sqrt{A_{ij}/d_{i}}$, then $S_{2}\left(x\right)=x^{T}L^{\left(n\right)}x$, which is denoted hereafter by $S_{2}^{\left(n\right)}\left(x\right)$ for the sake of clarity.

A relaxed definition of the local variation is proposed in [1,14] that uses the normalized adjacency matrix $\bar{A}={\left|\lambda_{\max}\right|}^{-1}A$ as a shift operator. This matrix guarantees that the energy of the shifted signal is not scaled up as ${∥\bar{A}x∥}_{2}/{∥x∥}_{2}\leq 1$. Then, the local variation at vertex i∈V is defined as
(9)$$||\nabla x_{i}^{\left(a\right)}\left|\right|:=\left|x_{i}-\sum_{j\in N_{i}}{\bar{A}}_{ij}x_{j}\right|$$

A smoothness measure is then given by the following *p*-Dirichlet form,
(10)$$S_{p}^{\left(a\right)}\left(x\right)=\frac{1}{p}\sum_{i=1}^{N}||\nabla x_{i}^{\left(a\right)}{\left|\right|}^{p}$$

With p=1, $S_{1}^{\left(a\right)}\left(x\right)={∥x-\bar{A}x∥}_{1}$ defines the global smoothness of the graph signal as the l1-distance between the graph signal and its shifted version. For p=2, $S_{2}^{\left(a\right)}\left(x\right)=\frac{1}{2}{∥x-\bar{A}x∥}_{2}^{2}$ is a measure proportional to the Euclidean distance between signals. In [14], it is also shown that graph frequency order based on the total variation (p=1) and the quadratic form (p=2) is equivalent.

In the next section, we use the random-walk normalized Laplacian matrix to propose a measure of smoothness of graph signals based on $S_{2}^{\left(a\right)}\left(x\right)$.

## 3. Random-Walk Normalized Laplacian Matrix for Frequency Analysis in Graphs

A random walk defined in G [25] is a path given by a sequence of *S* vertices $\left\{v_{1},\ldots ,v_{S}\right\}$, where $\left\{v_{t}\in V;\forall t=0,\ldots ,S-1\right\}$, obtained as a result of a stochastic process defined by a random walker that at time t≥0, is located at vertex $v_{t}=i$ and jumps to the next vertex $v_{t+1}=j$ with transition probability $p_{ij}P\left(v_{t+1}=j|v_{t}=i\right)=A_{ij}/d_{i}\;\forall j\in V$. Note that in graphs with self-loops, the random walker can stay in the same vertex. If the random walk is a Markovian stochastic process, i.e., only depends on the current state, then it is completely described by the transition matrix $P\in {\left[0,1\right]}^{N\times N}$, which can be computed as $PD^{-1}A$ and $P_{ij}=p_{ij}\;\forall i,j\in V$. The random-walk normalized Laplacian matrix is a normalized version of the Laplacian matrix defined as
(11)$$L^{\left(rw\right)}:=D^{-1}L=I-P$$
where I is the identity matrix. For undirected graphs, i.e., no direction is assigned to the edges so that (i,j)∈E→(j,i)∈E∀i,j∈V, matrices L and $L^{\left(n\right)}$ are symmetric and, therefore, they diagonalize with an orthonormal basis given by $L=V\Lambda V^{H}$ and $L^{\left(n\right)}=V^{\left(n\right)}\Lambda^{\left(n\right)}V^{(n)H}$, respectively. However, the random-walk Laplacian $L^{\left(rw\right)}$ diagonalizes as $L^{\left(rw\right)}=V^{\left(rw\right)}\Lambda^{\left(rw\right)}V^{(rw)-1}$ as it is not normal. Interestingly, diagonalization of $L^{\left(rw\right)}$ is guaranteed because it is similar to $L^{\left(n\right)}$ as $L^{\left(rw\right)}=D^{-1/2}L^{\left(n\right)}D^{+1/2}$.

### 3.1. Spectral Properties of the Random-Walk Laplacian

In undirected graphs, the eigenvalues of the Laplacian matrix are real and graph-dependent as their upper bound depends on the value of the highest vertex degree or max-degree, i.e.,
(12)$$0\leq \lambda_{1}\leq \cdots \leq \lambda_{N}\leq 2d_{\max}$$

With regards to $L^{\left(rw\right)}$ and $L^{\left(n\right)}$, they share the same set of eigenvalues as they are similar matrices. As opposed to ${\left\{\lambda_{i}\right\}}_{i=1}^{N}$ in (Equation 12), the upper bound of the eigenvalues of $L^{\left(rw\right)}$ and $L^{\left(n\right)}$ is graph independent, i.e.,
(13)$$0\leq \lambda_{1}^{\left(rw\right)}=\lambda_{1}^{\left(n\right)}\leq \lambda_{2}^{\left(rw\right)}=\lambda_{2}^{\left(n\right)}\leq \cdots \leq \lambda_{N}^{\left(rw\right)}=\lambda_{N}^{\left(n\right)}\leq 2,$$

This makes both $L^{\left(rw\right)}$ and $L^{\left(n\right)}$ better suited than **L** to study and compare graphs with different max-degree in the spectral domain. The eigenvectors of L and $L^{\left(rw\right)}$ with associated eigenvalue equal to 0 are both equal to an all ones vector. That is, both Laplacian matrices share the property that a constant or DC graph signal is built only with the zero frequency of the graph. However, this property is not satisfied by $L^{\left(n\right)}$ as $L^{\left(n\right)}1\neq 0$. Indeed, the interpretation that the eigenvalues of a Laplacian matrix is a measure of the smoothness of their eigenvectors across the graph is different for the three Laplacian matrices. It is not difficult to see that the *m*th eigenvalue of matrices L, $L^{\left(n\right)}$ and $L^{\left(rw\right)}$ of an undirected graph can be written, respectively, as
(14)$$\begin{array}{cc} \lambda_{m} & =S_{2}\left(v_{m}\right)=\frac{1}{2}\sum_{(i,j)\in E}A_{ij}{\left(v_{m,i}-v_{m,j}\right)}^{2} \end{array}$$
(15)$$\begin{array}{cc} \lambda_{m}^{\left(n\right)} & =S_{2}^{\left(n\right)}\left(v_{m}^{\left(n\right)}\right)=\frac{1}{2}\sum_{(i,j)\in E}A_{ij}{\left(\frac{v_{m,i}^{\left(n\right)}}{\sqrt{d_{i}}}-\frac{v_{m,j}^{\left(n\right)}}{\sqrt{d_{j}}}\right)}^{2} \end{array}$$
(16)$$\begin{array}{cc} \lambda_{m}^{\left(rw\right)} & =\frac{1}{2}\sum_{(i,j)\in E}A_{ij}\left(v_{m,i}^{\left(rw\right)}-v_{m,j}^{\left(rw\right)}\right)\left(\frac{v_{m,i}^{\left(rw\right)}}{\sqrt{d_{i}}}-\frac{v_{m,j}^{\left(rw\right)}}{\sqrt{d_{j}}}\right) \end{array}$$
where $v_{m,i},v_{m,i}^{\left(n\right)}$, and $v_{m,i}^{\left(rw\right)}$ denote the *i*th component of eigenvectors $v_{m},v_{m}^{\left(n\right)}$, and $v_{m}^{\left(rw\right)}$ for m=1,…,N, respectively. The eigenvalues of L depend on the squared difference between the entries of the associated eigenvector that correspond to connected vertices, and therefore render a measure of smoothness. However, the eigenvalues of $L^{\left(n\right)}$ depend on the eigenvector entries normalized by the square root of their associated vertex degree, and they do not measure smoothness unless $d_{i}=d_{j}\;\forall i,j\in V$. As for $L^{\left(rw\right)}$, when an eigenvector varies smoothly across the graph, expression (Equation 16) tends to
$$\begin{array}{c} \lim_{v_{m,i}^{\left(rw\right)}\to v_{m,j}^{\left(rw\right)}}\lambda_{m}^{\left(rw\right)}=\lim_{v_{m,i}^{\left(rw\right)}\to v_{m,j}^{\left(rw\right)}}\frac{1}{2}\sum_{(i,j)\in E}A_{ij}v_{m,i}^{\left(rw\right)}\frac{\sqrt{d_{j}}-\sqrt{d_{i}}}{\sqrt{d_{i}d_{j}}}\left(v_{m,i}^{\left(rw\right)}-v_{m,j}^{\left(rw\right)}\right) \end{array}$$

That is, similarly to **L**, the eigenvectors associated to small eigenvalues of $L^{\left(rw\right)}$ can be seen as graph signals that vary smoothly across the graph.

In summary, the eigenvalues of the random-walk Laplacian matrix are a good option to define the frequencies in a graph as they are bounded within a range that is independent of the graph max-degree, and the zero frequency corresponds to a DC graph signal. For an undirected graph, the eigenvalues of $L^{\left(rw\right)}$ are real and positive, and therefore they can be ordered and measure the smoothness using (Equation 8). However, in directed graphs the eigenvalues of $L^{\left(rw\right)}$ are complex and, therefore, they cannot be ordered. Therefore, in the next section we propose to use the quadratic form $S_{2}^{\left(a\right)}\left(x\right)$ in (Equation 10) replacing $\bar{A}$ by the transition matrix P, as a measure of smoothness and as a frequency ordering criterion that proves to be useful even in the complex case.

### 3.2. Quadratic Form with the Transition Matrix

According to the work in [14], the Euclidean distance between a graph signal $x\in R^{N}$ and its shifted version is a proper measure of global variation in (Equation 10). Using the transition matrix as shift operator, i.e., S=P, the quadratic form $S_{2}^{\left(a\right)}\left(x\right)$ becomes
(17)$$S_{2}^{\left(p\right)}\left(x\right)={∥x-Px∥}_{2}^{2}$$
where factor $\frac{1}{2}$ is omitted for simplicity. The smoothness of an eigenvector $v_{m}^{\left(rw\right)}$ is equal to
(18)$$S_{2}^{\left(p\right)}\left(v_{m}^{\left(rw\right)}\right)=\left|\right|v_{m}^{\left(rw\right)}-Pv_{m}^{\left(rw\right)}{\left|\right|}_{2}^{2}=\left|\right|L^{\left(rw\right)}v_{m}^{\left(rw\right)}{\left|\right|}_{2}^{2}={\left|\lambda_{m}^{\left(rw\right)}\right|}^{2}$$
as $v_{m}^{\left(rw\right)}$ is unit-norm. Let us consider two eigenvalues $\lambda_{m}^{\left(rw\right)}$ and $\lambda_{n}^{\left(rw\right)}$ with associated eigenvectors $v_{m}^{\left(rw\right)}$ and $v_{n}^{\left(rw\right)}$. If the eigenvalues satisfy $|\lambda_{m}^{\left(rw\right)}\left|>\right|\lambda_{n}^{\left(rw\right)}|$, then
(19)$$S_{2}^{\left(p\right)}\left(v_{m}^{\left(rw\right)}\right)-S_{2}^{\left(p\right)}\left(v_{n}^{\left(rw\right)}\right)=\left(|\lambda_{m}^{\left(rw\right)}{|}^{2}-{\left|\lambda_{m}^{\left(rw\right)}\right|}^{2}\right)>0$$

To summarize, (Equation 19) provides a criterion to order the eigenvalues of the random-walk Laplacian similar to that in [14] and allows us to interpret the eigenvalues of $L^{\left(rw\right)}$ as frequencies of the graph. The next section includes an example for a directed cyclic graph.

### 3.3. Prominent Example: Directed Cycle Graph

A discrete periodic signal may be regarded as a graph signal that evolves on a directed cycle graph. The adjacency matrix A of a directed cycle graph is non-symmetric but normal, so that the eigendecomposition of the adjacency matrix is $A=\frac{1}{N}F\Lambda^{\left(a\right)}F^{H}$, where the eigenvector matrix $F=\left[f_{1},f_{2},\ldots ,f_{N}\right]\in C^{N\times N}$ is the Fourier matrix and $\Lambda^{\left(a\right)}=\mathrm{diag}\left(e^{-j\omega_{1}},e^{-j\omega_{2}},\ldots ,e^{-j\omega_{N}}\right)$ is a diagonal matrix with the eigenvalues, where $\omega_{m}=\frac{2\pi (m-1)}{N}\;\forall m=1,\ldots ,N$. The *m*th eigenvector of the adjacency matrix is given by $f_{m}={\left[1,e^{j\omega_{m}1},\ldots ,e^{j\omega_{m}\left(N-1\right)}\right]}^{T}\in C^{N}$ and equals the unit-norm complex exponential used to compute the *m*th component of the DFT of a graph-signal at frequency $\omega_{m}$. The eigenvalues of the adjacency matrix are given by $\lambda_{m}^{\left(a\right)}=e^{-j\omega_{m}}$, shown in Figure 1.

In the directed cycle graph each vertex has unit degree, i.e., D=I, and the random-walk Laplacian becomes $L^{\left(rw\right)}=I-A=\frac{1}{N}F\left(I-\Lambda^{\left(a\right)}\right)F^{H}$, with eigenvalues equal to $\lambda_{m}^{\left(rw\right)}=1-e^{-j\omega_{m}}\;\forall m=1,\ldots ,N$. Then,
(20)$$S_{2}^{\left(rw\right)}\left(v_{m}^{\left(rw\right)}\right)=\left|\lambda_{m}^{\left(rw\right)}\right|=2\left|\sin\frac{\omega_{m}}{2}\right|$$

If the adjacency matrix were used as graph-shift operator to define the GFT, the phase of the eigenvalues of A would encode the frequency information of the graph as shown in Figure 1. This would not be in accordance with the well-known property that the maximum frequency of the DFT is located at ω=π and not at ω=2π. Instead, the 2-Dirichlet form in (Equation 20) measures the smoothness of an eigenvector of the random-walk Laplacian matrix as the Euclidean distance between 1 and the corresponding eigenvalue of the adjacency matrix $\lambda_{m}^{\left(a\right)}=e^{-j\omega_{m}}$. As seen in Figure 1, this distance is maximum for ω=π and it is the same for eigenvectors with the same frequency, a fact that is consistent with the interpretation of frequency in the DFT.

## 4. Frequency Analysis in Periodic Graphs

This section studies the relation between the eigenvalues of the random-walk Laplacian matrix of a graph and a periodic or replicated version. As seen in Section 3.2, the modulus of these eigenvalues is a measure of the variation of the eigenvectors across the graph and, therefore, the eigenvalues are well suited to be seen as frequencies of the graph. First, we define a periodic graph and study the spectrum of its random-walk Laplacian matrix compared to the spectrum of the basic graph. Interestingly, we show that when one graph is replicated, there is an upsampling effect in the frequency domain of the graph and, therefore, a frequency resolution increase.

### 4.1. Random-Walk Laplacian of Periodic Graphs

We refer to a periodic graph as a graph where a given pattern or *basic graph* is repeated several times across the graph. As a toy example, the graph in Figure 2a shows the basic graph given by G=(V,E) with N=5 vertices, while the one in Figure 2b is a periodic version of G with P=3 replicas and $N_{P}=P\cdot \left(N-1\right)+1=13$ vertices denoted by $G_{P}=\left(V_{P},E_{P}\right)$.

The Laplacian matrix of $G_{P}$ is denoted by $L_{P}$ and is built by sliding the Laplacian matrix of G, denoted by **L**, N−1 positions across its diagonal as many times as replicas of the basic graph includes $G_{P}$. Figure 3 shows the resulting Laplacian matrix of a periodic graph using P=2 replicas. Clearly, the entry $L_{NN}$ overlaps with $L_{11}$, meaning that the *N*th vertex of $G_{P}$ increases its degree with respect to the *N*th vertex of G by summing up the neighbors of vertex 1. If *P* replicas were present in the periodic graph, the vertices indexed by U={N,2N−1,3N−2,…,PN−P+1} would increase its degree by $|N_{1}|$. Without loss of generality, we assume that the vertex of the basic graph selected to concatenate the pattern is indexed by node *N*.

Interestingly, the random-walk Laplacian of the periodic graph given by $L_{P}^{\left(rw\right)}=D_{P}^{-1}L_{P}$ cannot be built by sliding the random-walk Laplacian of the basic graph $L^{\left(rw\right)}$ across the diagonal as the Laplacian matrix does. This is because the rows of $L_{P}$ corresponding to the overlapping nodes U would be divided by the new degree. Besides, note that the periodic graph is not equivalent to the Kronecker product graph of the basic graph with another graph [27].

### 4.2. Random-Walk Laplacian Eigenvalues of Periodic Graphs

We are now interested in studying the eigenvalues of the $L_{P}^{\left(rw\right)}$ in comparison with the eigenvalues $L^{\left(rw\right)}$. The next theorem establishes that the eigenvalues of $L^{\left(rw\right)}$ are also eigenvalues of $L_{P}^{\left(rw\right)}$.

**Theorem** **1.**
*Given a basic graph G=(V,E) with N vertices and random-walk Laplacian matrix $L^{\left(rw\right)}$, consider a periodic graph $G_{P}=\left(V_{P},E_{P}\right)$ with $N_{P}$ vertices and random-walk Laplacian matrix $L_{P}^{\left(rw\right)}$ built by concatenating P replicas of the basic graph at the set of vertices U={N,2N−1,3N−2,...,PN−P+1}. It holds that the eigenvalues of $L^{\left(rw\right)}$ denoted by $\left\{\lambda_{1}^{\left(rw\right)},\ldots ,\lambda_{N}\left(rw\right)\right\}$ are also eigenvalues of $L_{P}^{\left(rw\right)}$.*


**Proof**.We consider first P=2. Without loss of generality let us denote by $\lambda^{\left(rw\right)}$ any of the eigenvalues of $L^{\left(rw\right)}$. This eigenvalue satisfies that $\det(L^{\left(rw\right)}-\lambda^{\left(rw\right)}I)=0$, which means that the rows of $\left(L^{\left(rw\right)}-\lambda^{\left(rw\right)}I\right)$ denoted by $\left\{h_{1}^{T},\ldots ,h_{N}^{T}\right\}$ are linearly dependent. Let us now inspect $\left(L_{P}^{\left(rw\right)}-\lambda^{\left(rw\right)}I\right)$ for P=2.
(21)$$\begin{array}{cc} \; & L_{2}^{\left(rw\right)}-\lambda^{\left(rw\right)}I=\left[\begin{array}{ccc} h_{1}^{T} & \;| & 0^{T}\\ \ldots & \;| & \ldots \\ h_{N-1}^{T} & \;| & 0^{T}\\ \frac{d_{N}}{d_{N}+d_{1}}h_{N}^{T}\left(1:N-1\right) & \left(1-\lambda^{\left(rw\right)}\right) & \frac{d_{1}}{d_{N}+d_{1}}h_{1}^{T}\left(2:N\right)\\ 0^{T} & | & h_{2}^{T}\\ \ldots & |\; & \ldots \\ 0^{T} & |\; & h_{N}^{T} \end{array}\right] \end{array}$$
where 0 is a zero vector of length N−1, $d_{n}$ is the degree of the *n*th vertex of the basic graph G, and $h_{n}\left(l:m\right)$ is a vector built selecting from entry *l* to entry *m* of vector $h_{n}$.As $\left\{h_{1}^{T},\ldots ,h_{N}^{T}\right\}$ are linearly dependent, there exists a set of scalars ${\left\{\alpha_{n}\right\}}_{n=1}^{N}$, where at least one of them different from 0, such that $\sum_{n=1}^{N}\alpha_{n}h_{n}^{T}=0$. Two cases can be distinguished:
(i)Both $\alpha_{1},\alpha_{N}\neq 0$. In this case, there surely exists a linear combination of $\left\{h_{1}^{T},\ldots ,h_{N-1}^{T}\right\}$ equal to $-\frac{d_{N}}{d_{N}+d_{1}}h_{N}^{T}$ and another linear combination of $\left\{h_{2}^{T},\ldots ,h_{N}^{T}\right\}$ equal to $-\frac{d_{1}}{d_{N}+d_{1}}h_{1}^{T}$. Therefore, there exists a linear combination of the N−1 first and ending rows of $\left(L_{2}^{\left(rw\right)}-\lambda^{\left(rw\right)}I\right)$ in (Equation 21) equal to
(22)$$\left[-\frac{d_{N}}{d_{N}+d_{1}}h_{N}^{T}\left(1:N-1\right)|\beta |-\frac{d_{1}}{d_{N}+d_{1}}h_{1}^{T}\left(2:N\right)\right]$$
where the term in the middle is equal to
(23)$$\beta =-\frac{d_{N}\cdot h_{N}\left(N\right)+d_{1}\cdot h_{1}\left(1\right)}{d_{N}+d_{1}}$$
As per construction of $L_{2}^{\left(rw\right)}$ we have that $h_{1}\left(1\right)=h_{N}\left(N\right)=1-\lambda^{\left(rw\right)}$, and then β in (Equation 23) is also $-(1-\lambda^{\left(rw\right)})$. Therefore, (Equation 22) is equal to the *N*th row in (Equation 21), which means that the rows of $\left(L_{2}^{\left(rw\right)}-\lambda^{\left(rw\right)}I\right)$ are linearly dependent, $\det(L_{2}^{\left(rw\right)}-\lambda^{\left(rw\right)}I)=0$, and $\lambda^{\left(rw\right)}$ is an eigenvalue of $L_{2}^{\left(rw\right)}$ as well.(ii)At least one of $\left\{\alpha_{1},\alpha_{N}\right\}$ is equal to 0. In this case, there exists a linear combination of $\left\{h_{1}^{T},\ldots ,h_{N-1}^{T}\right\}$ (and/or $\left\{h_{2}^{T},\ldots ,h_{N}^{T}\right\}$) that is equal to $0^{T}$. This means that the first (and/or last) N−1 rows of the matrix in (Equation 21) are linearly dependent and, therefore, $\det(L_{2}^{\left(rw\right)}-\lambda^{\left(rw\right)}I)=0$ and $\lambda^{\left(rw\right)}$ is an eigenvalue of $L_{2}^{\left(rw\right)}$ as well. Note that $\det(L_{2}^{\left(rw\right)}-\lambda^{\left(rw\right)}I)=0$ when both $\alpha_{1}=\alpha_{N}=0$.

Following the same procedure, it is not difficult to see that $\lambda^{\left(rw\right)}$ is also an eigenvalue of $L_{P}^{\left(rw\right)}$∀P>2. □

**Remark 1**.*Theorem 1 also holds for the normalized Laplacian matrix as it has the same set of eigenvalues as the random-walk Laplacian matrix. Further, following a similar procedure, it can be proved that the eigenvalues of***L***are also eigenvalues of*$L_{P}$.


## 5. Numerical Results

This section includes numerical experiments that illustrate the advantage of using the random-walk Laplacian matrix as a graph shift operator for graph spectral analysis and universal graph filtering. Further, some results are included to support Theorem 1.

First of all, we present a simple visual example to show the spectral properties of $L^{\left(rw\right)}$ provided in Section 3.2. Figure 4 depicts the eigenvectors {$v_{1}^{\left(rw\right)}$, $v_{2}^{\left(rw\right)}$, $v_{6}^{\left(rw\right)}$, $v_{11}^{\left(rw\right)}$, $v_{51}^{\left(rw\right)}$, $v_{101}^{\left(rw\right)}$, $v_{501}^{\left(rw\right)}$, $v_{801}^{\left(rw\right)}$} of $L^{\left(rw\right)}$ for the GSPlogo graph [28] ordered according to the absolute value of the associated eigenvalue and represented as graph signals. The GSPlogo graph is an undirected connected graph with N=1130 nodes and |E|=3131 edges. Clearly, the smaller the absolute value of an eigenvalue is, the smoother the associated eigenvector varies across the graph. Furthermore, the eigenvector associated to $\lambda_{1}^{\left(rw\right)}=0$ is an all ones vector.

### 5.1. Random-Walk Laplacian for Graph Signal Denoising

Here, we use the random-walk Laplacian matrix as a graph-shift operator for the design of universal graph filters [29]. A universal graph filter is defined for a continuous range of λ’s rather than for the set of eigenvalues of a particular graph and, therefore, it can be used for graphs with different number of nodes or degrees. As the eigenvalues are $\lambda^{\left(rw\right)}\in \left[0,2\right]$, the random-walk Laplacian $L^{\left(rw\right)}$ is a suitable graph shift operator to design these filters.

An FIR graph filter of order *K* with graph-shift operator $L^{\left(rw\right)}$ is given by $H=\sum_{k=0}^{K}\phi_{k}\cdot {\left(L^{\left(rw\right)}\right)}^{k}$, and its frequency response by $h\left(\lambda_{m}^{\left(rw\right)}\right)=\sum_{k=0}^{K}\phi_{k}\cdot {\left(\lambda_{m}^{\left(rw\right)}\right)}^{k}$. The objective here is to design the coefficients $\left\{\phi_{k};k=0,\ldots ,K\right\}$ such that the frequency response of the FIR graph filter approximates an ideal low-pass filter given by
(24)$$\tilde{h}\left(\lambda \right)=\left\{\begin{array}{cc} 1 & 0\leq \lambda \leq \lambda_{c}\\ 0 & \lambda >\lambda_{c} \end{array}\right.$$
where $\lambda_{c}$ is the cut-off frequency. Then, the same set of coefficients can be used to filter a noisy signal defined on two different graphs.

Following the procedure in [29], the coefficients are found solving the least-squares problem:(25)$$\begin{array}{c} \left\{{\hat{\phi }}_{0},\ldots ,{\hat{\phi }}_{K}\right\}=\underset{\forall \phi_{1},\ldots ,\phi_{K}\in R}{arg min}\sum_{l=0}^{L-1}{\left(\tilde{h}\left(f_{l}\right)-\sum_{k=0}^{K}\phi_{k}\cdot f_{l}^{k}\right)}^{2} \end{array}$$
where $\{f_{l}=\frac{l\cdot 2}{L}$ for {l=0,…,L−1}. That is, the coefficients $\left\{{\hat{\phi }}_{0},\ldots ,{\hat{\phi }}_{K}\right\}$ minimize the mean square error between *L* samples of the ideal low-pass frequency response $\tilde{h}\left(\lambda \right)$ taken uniformly in the range of frequencies [0,2], and the frequency response of the FIR graph filter at those frequency samples. As the error depends linearly on the coefficients, the solution to (Equation 25) is
(26)$$\hat{\phi }=\Xi^{†}\cdot \tilde{h}$$
where $\hat{\phi }={\left[{\hat{\phi }}_{0}\cdots {\hat{\phi }}_{K}\right]}^{T}$, $\tilde{h}={\left[\tilde{h}\left(f_{0}\right)\cdots \tilde{h}\left(f_{L-1}\right)\right]}^{T}$ and Ξ is the Vandermonde matrix
(27)$$\Xi =\left[\begin{array}{ccccc} 1 & f_{0} & f_{0}^{2} & \cdots & f_{0}^{K}\\ 1 & f_{1} & f_{1} & \cdots & f_{1}^{K}\\ \vdots & \vdots & \vdots & \ddots & \vdots \\ 1 & f_{L-1} & f_{L-1}^{2} & \cdots & f_{L-1}^{K} \end{array}\right]$$

In this experiment, we consider the GSPlogo graph with N=1130 (top graph in Figure 5) and a random geometric graph with N=800 (bottom graph in Figure 5). The number of frequency samples is L=100 and $\lambda_{c}$ = 5 $\frac{}{L}$ so that $\tilde{h}$$={\left[1\;1\;1\;0\cdots 0\right]}^{T}$. An FIR graph filter $H=\sum_{k=0}^{K}{\hat{\phi }}_{k}\cdot {\left(L^{\left(rw\right)}\right)}^{k}$ of order K=8 is computed for each graph using the same set of coefficients given by (Equation 26). This filter is used to denoise a graph signal defined on each graph and equal to $x=v_{2}^{\left(rw\right)}+n$, where $v_{2}^{\left(rw\right)}$ is the second eigenvector of the corresponding $L^{\left(rw\right)}$ and n is a white Gaussian noise $n∼N(0,\frac{1}{N}I)$. Results are shown in Figure 5. Signals on the left hand side are the eigenvectors $v_{2}^{\left(rw\right)}$, on the center are the noisy signals x, and on the right the filtered signals given by y=Hx. Clearly, the designed graph system filters out the noise as the filtered graph signal resembles the original one $v_{2}^{\left(rw\right)}$ on the left.

### 5.2. Frequency Aliasing of Replicated Graphs

In this section, we support the results of Theorem 1 through simulations, and compare the eigenvalues of the three Laplacian matrices when a basic graph is replicated. Given the basic graph in Figure 2a, the eigenvalues of $L^{\left(rw\right)}$ and $L^{\left(n\right)}$ are equal to {0,0.85,1.25,1.33,1.56}, and of **L** are {0,2,4,5,5}. The eigenvalues of $L_{2}^{\left(rw\right)}$ and $L_{2}^{\left(n\right)}$ are {0,0.18,0.85,1.09,1.25,1.33,1.33,1.4,1.56}, and of $L_{2}$ are {0,0.6,2,4,4,4.18,5,5,7.22}. Finally, the eigenvalues of $L_{3}^{\left(rw\right)}$ and $L_{3}^{\left(n\right)}$ are {0,0.1,0.25,0.85,1.04,1.12,1.25,1.33,1.33,1.33,1.36,1.46,1.56}, and of $L_{3}$ are {0,0.33,0.84,2,4,4,4,4.07,4.46,5,5,6.54,7.74}. For clarity, the eigenvalues of $L_{P}$ for P={1,2,3} are plotted in Figure 6 and Figure 7 for $L_{P}^{\left(rw\right)}$ and $L_{P}^{\left(n\right)}$. Clearly, Theorem 1 holds for the three Laplacian matrices.

Interestingly, in Figure 7 we also observe that a periodic graph with *P* replicas inserts P−1 additional eigenvalues between two consecutive eigenvalues of $L^{\left(rw\right)}$. This is a kind of nonuniform upsampling in the frequency domain of the graph, and translates into an increase in the frequency resolution of the graph. Remarkably, as it can be seen in Figure 6, this is not the case for the eigenvalues of **L**, as some of the new eigenvalues of $L_{P}$ are outside the range of eigenvalues of **L**. On the other hand, as $\lambda^{\left(n\right)}=\lambda^{\left(rw\right)}$, the normalized Laplacian matrix $L^{\left(n\right)}$ also presents the same upsampling in the frequency domain when a graph is replicated. However, as discussed in Section 3.1 and evidenced by (Equation 16), the eigenvalues of the normalized Laplacian matrix do not properly measure smoothness of their associated eigenvectors as those of the random-walk Laplacian do.

## 6. Conclusions

A criterion for frequency ordering in both directed and undirected graphs using the modulus of the eigenvalues of the random-walk normalized Laplacian matrix is proposed. Indeed, the random-walk Laplacian matrix proves to be a graph-shift operator suitable for both graph spectral analysis and graph filter design in undirected graphs due to the graph-independent bounds of its eigenvalues. We also provide some insights into the relation between the eigenvalues of the random-walk Laplacian of a periodic graph and those of the basic graph used to build the periodic one. First, we prove that the eigenvalues of the basic graph are also eigenvalues of the periodic graph. Numerical results further show that the additional eigenvalues of the periodic graph appear to upsample those of the basic graph, in a similar way as zero-padding in DSP albeit not evenly spaced. The spectral properties of periodic graphs open several interesting research venues such as interpolation filter design for graph signal filtering, or graph pooling methods for graph CNNs.

## Figures and Tables

**Figure 1 sensors-21-01275-f001:**
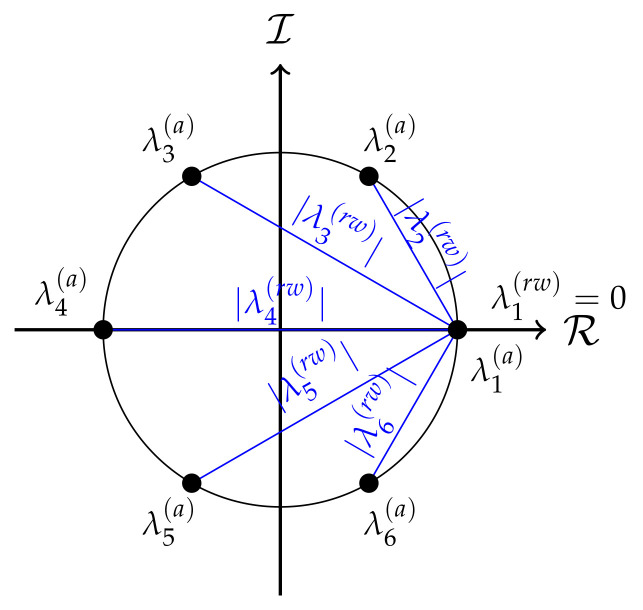
Eigenvalues of A given by $\left\{\lambda_{m}^{\left(a\right)};\forall m\right\}$ and modulus of the eigenvalues of $L^{\left(rw\right)}$ given by $\left\{\lambda_{m}^{\left(a\right)};\forall m\right\}$ in blue, of a directed cyclic graph with N=6 vertices.

**Figure 2 sensors-21-01275-f002:**
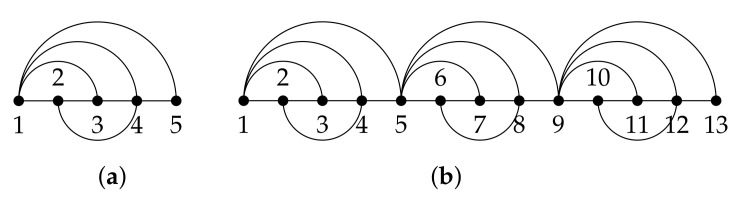
(**a**) Basic graph G=(V,E) [26]. (**b**) Periodic graph with *P* = 3 replicas $G_{P}=\left(V_{P},E_{P}\right)$.

**Figure 3 sensors-21-01275-f003:**
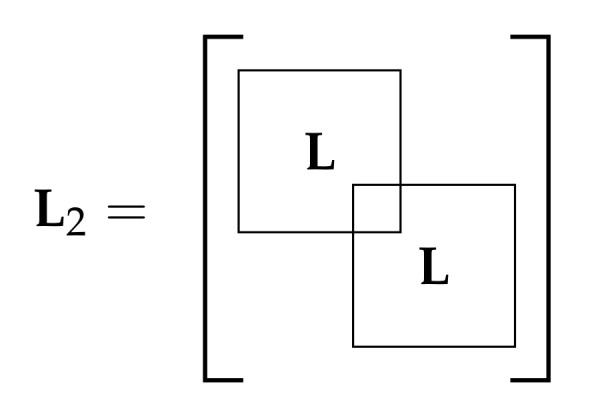
Laplacian matrix of a periodic graph $G_{P}$ with P=2.

**Figure 4 sensors-21-01275-f004:**
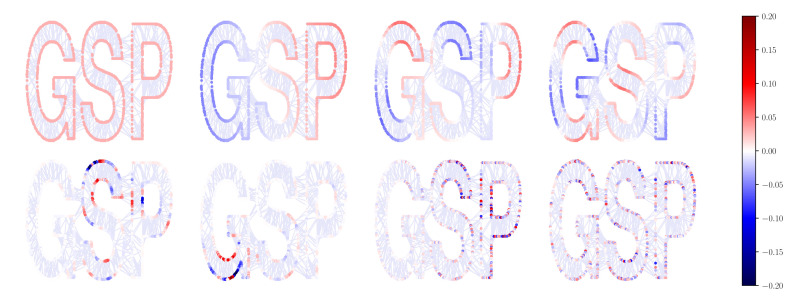
Eigenvectors of $L^{\left(rw\right)}$ for the GSPlogo graph corresponding to eigenvalues $\lambda_{1}^{\left(rw\right)}$, $\lambda_{2}^{\left(rw\right)}$, $\lambda_{6}^{\left(rw\right)}$, $\lambda_{11}^{\left(rw\right)}$, $\lambda_{51}^{\left(rw\right)}$, $\lambda_{101}^{\left(rw\right)}$, $\lambda_{501}^{\left(rw\right)}$, and $\lambda_{801}^{\left(rw\right)}$.

**Figure 5 sensors-21-01275-f005:**
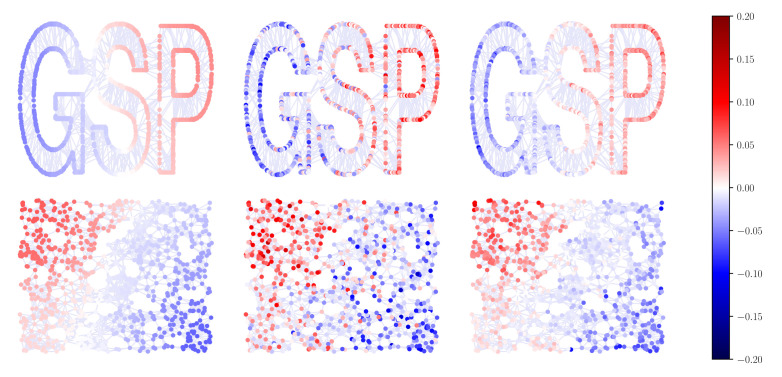
**Graphs**—*Top*: GSPlogo, N=1130 nodes. *Bottom*: Random graph, N=800 nodes. **Signals**: *Left*: Graph signals without noise. *Center*: Noisy graph signal *Right*: Graph signal filtered with the graph filter $H=\sum_{k=0}^{K}{\hat{\phi }}_{k}\cdot {\left(L^{\left(rw\right)}\right)}^{k}$.

**Figure 6 sensors-21-01275-f006:**
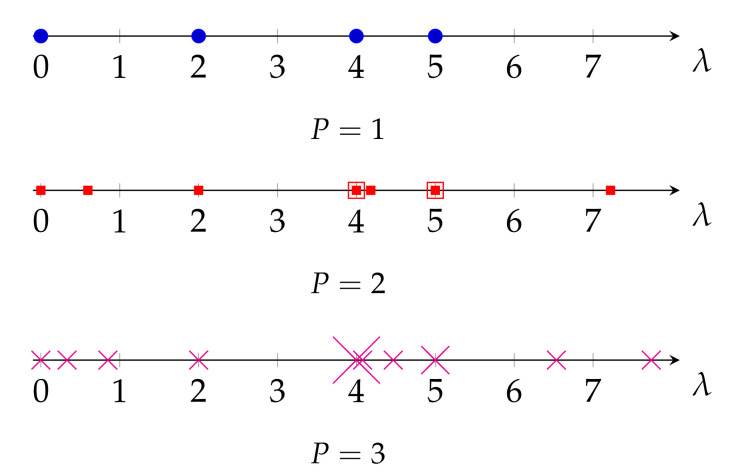
Eigenvalues of **L**_*p*_ for the graph in Figure 2b for *P* = {1, 2, 3}.

**Figure 7 sensors-21-01275-f007:**
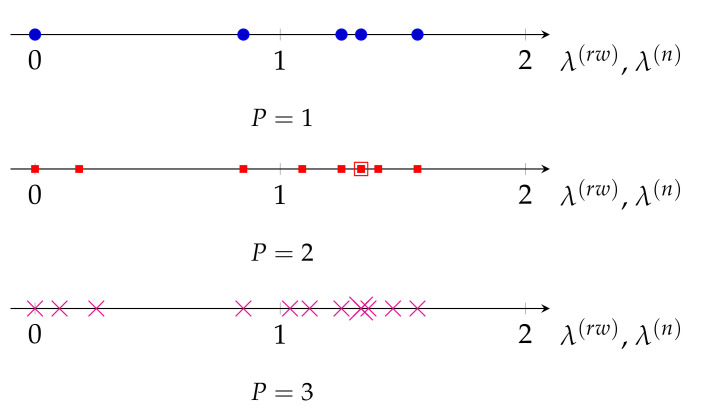
Eigenvalues of $L_{P}^{\left(rw\right)}$ and $L_{P}^{\left(n\right)}$ for the graph in Figure 2b for *P* = {1, 2, 3}.

## Data Availability

The GSPlogo graph and the random graph used in Section 5 were generated using the *GSPBox* toolbox. This toolbox is available in a publicly accessible GitHub repository: https://github.com/epfl-lts2/gspbox.

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
