# Peer review of "Random-Walk Laplacian for Frequency Analysis in Periodic Graphs"

_sensors, 2021, doi:10.3390/s21041275_

Round 1
Reviewer 1 Report
It is interesting to analyze the spectrum of random-walk Laplacian, while it is presented without clarifying the contribution of the theoretical finding in Theorem 1. Overall, the contribution is not clearly clarified. The remainder comments are below.
- Line 121, "the random-walk Laplacian diagonalizes...", since the random-walk Laplacian is asymmetric, diagonalization is not ensured.
2. The numerical examples are far from sufficient to illustrate the theoretical finding. There is no graph signal processing application included. The authors should conduct examples to recover the merit of using the spectrum of the random-walk Laplacian.
3. The function of the so-called upsampling of the replicated graphs is not clarified in section 5.
Reviewer 2 Report
The manuscript is well motivated and clearly written. The results are correct. In particular, I am a pure mathematician and have no experience in practical applications, but the results shown have been very pleasing to me and I find them novel and interesting.
I recommend the publication of the article, after a review of the English writing.
Round 2
Reviewer 1 Report
It seems that the authors have addressed my comments. Overall, the revised paper is well organization. I have an additional comment: what is the motivation of analysis the eigenvalues distribution of the replicated graph constructed by duplicating basic graphs? Is there any connection to the Kronecker product graph? The authors should provide the motivation in the paper.Author Response
Dear Reviewer,
We have thoroughly reviewed the English usage of the manuscript. Thank you for your comments and please see the attachment for our reply.
